# Novel Chemicals Derived from Tadalafil Exhibit PRMT5 Inhibition and Promising Activities against Breast Cancer

**DOI:** 10.3390/ijms23094806

**Published:** 2022-04-27

**Authors:** Ziyan Yang, Tian Xiao, Zezhi Li, Jian Zhang, Suning Chen

**Affiliations:** 1State Key Laboratory of Cancer Biology, Department of Biochemistry and Molecular Biology, Fourth Military Medical University, Xi’an 710032, China; yangziyan93@163.com (Z.Y.); bioxiaotian92@163.com (T.X.); 2Department of Pharmacy, School of Chemistry & Pharmacy, Northwest A&F University, Xianyang 712100, China; lizezhi3399@163.com; 3Department of Pharmacy, Fourth Military Medical University, Xi’an 710032, China

**Keywords:** breast cancer, tadalafil, PRMT5 inhibitors, chemotherapy drugs, olaparib

## Abstract

Breast cancer seriously endangers women’s health worldwide. Protein arginine methyltransferase 5 (PRMT5) is highly expressed in breast cancer and represents a potential druggable target for breast cancer treatment. However, because the currently available clinical PRMT5 inhibitors are relatively limited, there is an urgent need to develop new PRMT5 inhibitors. Our team previously found that the FDA-approved drug tadalafil can act as a PRMT5 inhibitor and enhance the sensitivity of breast cancer patients to doxorubicin treatment. To further improve the binding specificity of tadalafil to PRMT5, we chemically modified tadalafil, and designed three compounds, A, B, and C, based on the PRMT5 protein structure. These three compounds could bind to PRMT5 through different binding modes and inhibit histone arginine methylation. They arrested the proliferation and triggered the apoptosis of breast cancer cells in vitro and also promoted the antitumor effects of the chemotherapy drugs cisplatin, doxorubicin, and olaparib in combination regimens. Among them, compound A possessed the highest potency. Finally, the anti-breast cancer effects of PRMT5 inhibitor A and its ability to enhance chemosensitivity were further verified in a xenograft mouse model. These results indicate that the new PRMT5 inhibitors A, B, and C may be potential candidates for breast cancer treatment.

## 1. Introduction

Breast cancer accounts for approximately 30% of female cancers and exhibits a mortality-to-incidence ratio of 15% [1]. Approximately 10% of breast cancer patients have a family history or genetic predisposition. The most common germline mutations are located in the *BRCA1* and *BRCA2* genes, and these mutations indicate an average cumulative lifetime risk of approximately 70% [2,3,4]. At present, breast cancer is mainly treated by radiotherapy, chemotherapy, surgery, and targeted drugs according to the disease stage [4]. Chemotherapy plays a pivotal role in treatment. However, in the course of clinical treatment, an increasing number of breast cancer patients are less sensitive to anthracycline chemotherapeutics, such as doxorubicin, leading to drug resistance and reduced therapeutic effects [5,6,7].

Protein arginine methyltransferase 5 (PRMT5) belongs to type II arginine methyltransferases and primarily acts by catalyzing the symmetrical dimethylation of arginine residues in targeted proteins [8,9]. PRMT5 forms large complexes with WD repeat-containing protein methylosome protein 50 (MEP50) [10,11], and the highly conserved methyltransferase (Mtase) domain of PRMT5 contributes to *S*-adenosyl-L-methionine (SAM) binding and catalysis [12,13]. PRMT5 is overexpressed or hyperactive in a number of tumor types and promotes tumor progression, including breast cancer, acute myeloid leukemia, lung, ovarian, glioblastoma, and prostate cancer [14,15,16,17,18,19,20,21]. The inhibition of PRMT5 in tumor tissues can trigger DNA damage, defects in DNA repair, and activation of the p53 signaling pathway, leading to cell cycle arrest and cell death [8,9,22,23]. Recently, it was demonstrated that PRMT5-mediated arginine methylation could also activate the AKT signaling pathway in tumorigenesis [24]. In breast cancer cells, PRMT5 functions in various mechanisms. PRMT5 methylates KLF5 to prevent its phosphorylation, and thus promotes cell proliferation [18]. In addition, PRMT5 epigenetically silences DKK1 and DKK3, resulting in up-regulation of Wnt/β-catenin proliferative signaling [25]. Previous studies from our group also demonstrated that PRMT5 could reduce doxorubicin sensitivity in the treatment of breast cancer by inhibiting RNA m6A methylation [26,27].

Owing to the important role of PRMT5 in tumors, research into its inhibitors has generated intense interest. The inhibitors reported thus far, such as SAM-competitive inhibitors, ARG-competitive inhibitors, and allosteric inhibitors, suppress PRMT5 activities in different modes [28,29]. SAM, the cofactor that donates methyl group, and ARG, the cofactor that accepts methyl group, occupy distinct binding pockets [8]. Furthermore, allosteric inhibitors abrogate the canonical binding sites of PRMT5, such as compound 1a [8,29]. GSK3326595, which directly interacts with the departing methyl group of SAM and functions as an ARG-competitive inhibitor, is the first PRMT5 inhibitor tested in patients, and its phase I trial exhibited satisfactory clinical results. In addition, JNJ-64619178 and PF-06939999 were tested in phase I clinical trials [8,9,30]. However, to our knowledge, no approved PRMT5 inhibitors have been available for clinical use so far. This is a prominent challenge for targeting PRMT5 in breast cancer treatment. Therefore, finding suitable PRMT5 inhibitors to improve the sensitivity of chemotherapy drugs is of great significance for clinical breast cancer treatment.

Tadalafil was originally used to treat erectile dysfunction and pulmonary arterial hypertension [31,32,33]. Our group previously identified tadalafil as a new PRMT5 inhibitor that effectively improves the sensitivity of tumor cells to doxorubicin and other chemotherapeutic drugs in breast cancer, providing new insights into breast cancer treatment [27]. To further improve the binding specificity of tadalafil and PRMT5, we synthesized three chemicals based on the structural modification of tadalafil in this study and then tested their antitumor effects in vitro and in vivo when used alone or combined with chemotherapy drugs.

## 2. Results

### 2.1. Drug Design of Chemicals A, B, and C

The C-terminus of PRMT5 has a Mtase domain, which binds with SAM to play a catalytic role [13,34]. Given its core structure and hydrophobicity, the mechanism of tadalafil may be similar to SAM binding domain inhibitors [12,35,36,37].

First, Sybyl-X 2.0 software was used to analyze the molecular docking ability of some PRMT5 inhibitors that have been reported with PRMT5. The results showed that the binding affinity of tadalafil for PRMT5 was not as high as that of inhibitors, such as EPZ015666 (Figure 1A) [9]. Since our research group has previously verified that the FDA-approved drug tadalafil can be used as a PRMT5 inhibitor [27], we aimed to further enhance the ability of tadalafil to bind to PRMT5.

Therefore, we modified the structure of tadalafil as follows. First, groups containing nitrogen, oxygen, sulfur, and other heteroatoms were installed on the three relatively independent parts of tadalafil (the piperazine, indole, and benzene rings connected to the parent ring) to facilitate the formation of more hydrogen bonds between tadalafil and other proteins. Second, different alkyl substituents were introduced to enable tadalafil to interact with other molecules through hydrophobic interactions or aromatic groups. Third, the chirality of the chiral center was altered. Fourth, we introduced the above changes into tadalafil at the same time and explored the combined influence of these changes on the binding ability of tadalafil and other proteins.

Subsequently, molecular docking analysis was performed again on the molecular structure after modification. The three highest-scoring compounds were screened out of the 90 compounds, named A, B, and C. All three compounds were synthesized with a higher molecular docking score than tadalafil (Figure 1B).

### 2.2. Compounds A, B, and C Bind to the PRMT5 Protein through Different Binding Modes and Inhibit Histone Arginine Methylation

To predict the binding modes of the three inhibitors to the PRMT5 protein, chemicals A, B, and C were used to dock to the ARG binding site, SAM binding site, and allosteric site of PRMT5 separately (Figure 2A) [28,29]. The results showed that the skeleton structure (the four rings) of compound A binds to the SAM binding site of PRMT5, whereas the bromophenyl moiety of compound A extends to the ARG binding site (Figure 2B,C). Compound A binds with PRMT5 mainly through polar and hydrophobic interactions. The interacting polar residues involved Thr323, Tyr324, Glu328, Lys333, Tyr334, Glu435, Glu444, and Ser578, and the interacting hydrophobic residues involved Pro314, Leu319, Phe327, Pro370, Leu436, and Trp579 of PRMT5 (Figure 2D). Among these residues, Phe327, Glu435, Glu444, Ser578, and Trp579 of PRMT5 are also involved in interacting with the substrate ARG, which indicates that compound A might interfere with both SAM and ARG binding to human PRMT5 (Figure 2D). 

The molecular docking results suggest that compound B binds to an allosteric site of PRMT5 (Appendix A). The binding interactions are mainly hydrophobic interactions, and the involved residues include Leu315, Leu437, Pro447, Phe471, Leu472, Val503, Phe519, Trp579, Phe584, Phe555, Ile567, Phe580, Ile582, Phe584, and Phe602 (Appendix A). The dimethoxyphenyl group of compound B also forms an arene-H interaction with Ile582. In addition, compound B also forms polar interactions with Asn443, Glu444, Ser446, Tyr468, Ser470, Gly553, Tyr554, and Tyr613 of PRMT5 (Appendix A).

The binding mode of compound C with PRMT5 is similar to that of compound B. The molecular docking results suggest that compound C binds to an allosteric site of PRMT5 (Appendix A). Compound C binds with PRMT5 mainly through hydrophobic interactions, and the involved residues include Leu315, Leu437, Pro447, Phe471, Leu472, Val503, Phe519, Phe584, Phe555, Ile567, Phe580, and Ile582 (Appendix A). In addition, compound C also forms polar interactions with Asn443, Glu444, Ser446, Tyr468, Ser470, Gly553, Tyr554, and Tyr613 of PRMT5 (Appendix A). 

To gain further insight into the binding specificity of the three above-mentioned compounds, we used Surface Plasmon Resonance (SPR) to investigate the binding affinity of chemicals A, B, and C to the PRMT5/MEP50 complex. The binding of chemicals A, B, and C to the PRMT5/MEP50 complex was dose dependent, as indicated by the fast association-dissociation process, and the response units at equilibrium against the PRMT5/MEP50 complex were plotted. These results verify that the three compounds could bind to PRMT5 (Figure 3A).

PRMT5 could catalyze the symmetric dimethylation of histones H2AR3, H3R2, H3R8, and H4R3 [16,38,39], so we added compounds A, B, and C and tadalafil at 50, 100, and 150 μM to MDA-MB-231 triple-negative breast cancer cells [40] and performed western blotting to detect the total protein levels of H4R3me2s and H3R8me2s 48 h later. Compared with the control, compounds A, B, and C and tadalafil reduced the total levels of H4R3me2s and H3R8me2s in the cells in a dose-dependent manner. Among them, compound A exerted the strongest effect (Figure 3B). The above results indicate that these three compounds could bind to PRMT5 through different binding modes and inhibit histone arginine methylation, thus representing new PRMT5 inhibitors.

### 2.3. Compounds A, B, and C Inhibit Breast Cancer Cell Proliferation

Since compounds A, B, and C have inhibitory effects on PRMT5, we analyzed the effects of the three compounds on the proliferation of breast cancer cells in comparison with that of tadalafil. First, we used CCK-8 cell proliferation and toxicity testing experiments to observe the effects of compounds A, B, and C on MDA-MB-231 cell viability. At a concentration of 75 μM or more, compound A was better than the other two compounds, and the tendency to inhibit cell viability was stronger. Under 150 μM culture conditions, the gap between compound A and the other two compounds was the most obvious. Therefore, we performed a statistical analysis of cell viability at this concentration (Figure 4A). The CCK-8 results for ER-positive MCF-7 breast cancer cells (Figure 4B) and BRCA1-mutant HCC1937 breast cancer cells (Figure 4C) were similar to those for MDA-MB-231 cells [41,42].

Next, we used EdU incorporation experiments to further clarify the effects of the three compounds and tadalafil on the proliferation of MDA-MB-231 cells. In the CCK-8 cell proliferation and toxicity test, the difference in efficacy between the compounds at a concentration of 150 μM was the most significant. Therefore, this concentration was selected for detection in the EdU incorporation experiment. Similar to the results of the CCK-8 experiment, the three compounds showed varying degrees of inhibition of nucleotide incorporation during DNA replication, but compound A had the strongest inhibitory effect on nucleotide incorporation during DNA replication (Figure 5).

In summary, compounds A, B, and C could inhibit the proliferation of breast cancer cell lines with different degrees of malignancy when used alone, and compound A had a more pronounced effect than tadalafil.

### 2.4. Compounds A, B, and C Promote Breast Cancer Cell Apoptosis

Since knocking down PRMT5 in tumor cells can not only inhibit tumor cell proliferation but can also increase cell apoptosis [26], we further analyzed the effects of compounds A, B, and C on MDA-MB-231 cell apoptosis. The MDA-MB-231 cell line was treated with 150 μM compounds A, B, C, and tadalafil, and the changes in cell survival were analyzed by annexin V/propidium iodide (PI) double staining. After 48 h of drug treatment, the three compounds and tadalafil all promoted cell apoptosis. Among them, compound A had a stronger ability to promote apoptosis of cells than other compounds, as the compound A group had the lowest proportion of living cells and the highest proportion of annexin V^+^ PI^−^ and annexin V^+^ PI^+^ cells (Figure 6). The apoptosis effect in MCF7 cells was similar to that in MDA-MB-231 cells (Figure 7). Overall, the addition of compound A significantly inhibited cell survival and promoted cell apoptosis, and its efficacy was stronger than that of tadalafil and the other two compounds.

### 2.5. Compounds A, B, and C Promote the Antitumor Effect of Chemotherapeutics In Vitro

According to our previous work, tadalafil can enhance the sensitivity of doxorubicin to breast cancer treatment [27]. First, we analyzed the effects of compounds A, B, C, and tadalafil in combination with doxorubicin on MDA-MB-231 cell proliferation. In the CCK-8 experiment, MDA-MB-231 cells were treated with compounds A, B, C, and tadalafil at a concentration of 150 μM, and a concentration gradient of doxorubicin treatment was established to observe the effect of the compound and doxorubicin on cell viability. The results showed that when the concentration of doxorubicin treatment was 0.2 μg/mL, compounds A, B, C, and tadalafil could further enhance the inhibition of doxorubicin on cell viability, but compound A combined with doxorubicin had a more obvious effect on the inhibition of cell viability (Figure 8A). Moreover, we found that treatment with A, B, C, and tadalafil at higher concentrations of doxorubicin no longer enhanced the effect of doxorubicin, indicating that compounds A, B, C, and tadalafil can enhance the inhibitory effect of doxorubicin on cell proliferation within a certain concentration range (Figure 8A). Similar results were observed in MCF-7 cells (Figure 8B).

The clinical efficacy of cisplatin is dose-dependent, and high-dose cisplatin has greater toxicity and side effects for patients than low-dose cisplatin [43,44]. Therefore, we attempted to explore whether compounds A, B, C, and tadalafil can promote the antitumor effect of cisplatin, thereby reducing the dosage of cisplatin needed for clinical application. In MDA-MB-231 cells and MCF-7 cells, with 16 μM cisplatin treatment, compounds A, B, C, and tadalafil further promoted the inhibitory effect of cisplatin on cell proliferation, and the combined use of compound A and cisplatin had the most obvious inhibitory effect on cell viability (Figure 9).

The PARP inhibitor olaparib has a more obvious effect on patients with BRCA1 deficiency in the treatment of breast cancer, but patients with BRCA1 deficiency account for only a part of the population [45,46,47]. This is one of the constraints in the clinical application of PARP inhibitors. Previous results from our group showed that PRMT5 inhibition in breast cancer cells can promote the m6A modification of BRCA1 mRNA and reduce its stability [27]. Therefore, we speculated that the combination of compounds A, B, C, and tadalafil could promote the efficacy of the PARP inhibitor olaparib against breast cancer cells. In MDA-MB-231 and MCF-7 cells, the CCK-8 proliferation and cytotoxicity test showed that when the concentration of compounds A, B, C, and tadalafil was set to 150 μM, all the compounds could enhance the inhibitory effect of olaparib on cell viability, and compound A had the most pronounced effect (Figure 10).

In addition, we analyzed the effects of compound A and tadalafil on MDA-MB-231 cell apoptosis when combined with doxorubicin by FACS. Combined treatment with tadalafil and compound A, compared with doxorubicin alone, significantly reduced the proportion of living cells, and the proportion of cells in early apoptosis and late apoptosis was also significantly increased. Instead, the PI^+^ necrotic cell population did not change after compound A treatment. Overall, the combined use of compound A and doxorubicin significantly inhibited cell survival and promoted cell apoptosis, and the effect was stronger than that of doxorubicin alone and that of tadalafil and doxorubicin combined (Figure 11).

### 2.6. Compounds A, B, and C Promote the Antitumor Effects of Chemotherapeutics in Xenograft Tumor Models

To further confirm the antitumor effect of the novel PRMT5 inhibitor in vivo, we subcutaneously inoculated nude mice with MDA-MB-231 cells, and the mice were treated with chemical A in combination with doxorubicin, cisplatin, and olaparib. Consistent with the results from the in vitro experiments, the use of compound A alone slowed the growth of tumors (Figure 12). This finding differs from that obtained in our previous work, in which mice treated with tadalafil alone did not exhibit tumor growth suppression in the MDA-MB-231 mouse xenograft model [27]. The use of compound A in combination with chemotherapeutics further enhanced the efficacy of the chemotherapeutics (Figure 12). We also confirmed that the protein level of H3R8me2s was downregulated in tumors treated with compound A and that cleaved PARP was upregulated in the olaparib-treated group (Appendix A). 

Furthermore, the results of Ki67 immunohistochemistry showed that when compound A was used in combination with chemotherapy drugs, the proliferation ability of tumor cells was further inhibited (Figure 13A,C) [48]. Additionally, the expression of γH2A.X in tumor tissues increased, suggesting increased levels of DNA damage (Figure 13B,C) [49]. In summary, compound A can enhance the antitumor effects of doxorubicin, cisplatin, and olaparib in vivo.

## 3. Discussion

Breast cancer is an important disease endangering women’s health [1,4]. In the treatment of breast cancer, resistance to chemotherapeutic drugs, such as doxorubicin, has become a key factor affecting the treatment response [7,50,51]. PRMT5 is involved in the methylation modification of proteins, which can regulate DNA damage and repair, and plays a critical role in cell survival and growth [8,9,14]. Additionally, it can directly affect the therapeutic effect of doxorubicin [26]. However, research on PRMT5 inhibitors is in the preliminary stage, and new inhibitors require further development [8,12,36]. Our previous study showed that tadalafil can act as a PRMT5 inhibitor [27], but its binding ability to PRMT5 still needs to be improved. In this study, three new PRMT5 inhibitors were designed and synthesized based on tadalafil, and the antitumor effects of the new PRMT5 inhibitors alone and combined with chemotherapy drugs were evaluated in vitro and in vivo.

First, through molecular docking experiments, we modified the structure of tadalafil and obtained chemicals A, B, and C, which have a stronger binding ability to PRMT5. The SPR experiments confirmed that compounds A, B, and C can bind to PRMT5, and the molecular docking results suggested that the three compounds could bind to PRMT5 through different binding modes. Moreover, the new compounds decreased the histone modification levels of H4R3me2s and H3R8me2s in breast cancer cells. Then, we verified the proliferation-inhibiting and apoptosis-inducing effects of chemicals A, B, and C in breast cancer cells. Among them, compound A had the strongest antitumor effect when used alone. These results confirm that the newly synthesized compounds can act as novel PRMT5 inhibitors.

This study has its own strengths demonstrated in several aspects. First, based on the restructured tadalafil, we screened out more potent drug compound A, which showed a promising antitumor effect. Second, compound A could facilitate chemotherapeutic efficacy. Chemotherapy is currently one of the main clinical treatments for breast cancer, but the dose dependence and side effects of various chemotherapeutic drugs restrict them from wider application [7,50,52,53]. Compounds A, B, and C can promote the antitumor effect of chemotherapeutics, such as doxorubicin, cisplatin, and olaparib. Among these, compound A was the most effective in this context. Finally, it is worth noting that the PARP inhibitor olaparib has a more favorable effect on BRCA1-mutated patients in the clinical treatment of breast cancer, but the BRCA1-mutated patients account for only a portion of the population. Encouragingly, compounds A, B, and C could not only enhance the antitumor effect of olaparib in BRCA1-mutated HCC1937 cells but also function in BRCA1-wildtype cells, such as MDA-MB-231 and MCF-7 cells. This is of great significance for the expanded clinical application of PARP inhibitors [54,55].

However, our current study has several limitations to be acknowledged. The three new PRMT5 inhibitors have higher working concentrations than the commercially available PRMT5 inhibitors. Nevertheless, PRMT5^−/−^ mice exhibited embryonic death at E6.5 due to the abrogation of pluripotent cells in blastocysts, suggesting that PRMT5 plays a critical role in development [56]. For this reason, although commercially available PRMT5 inhibitors may possess a higher affinity to PRMT5, their corresponding toxicity and side effects cannot be ignored [12]. Taken together, inhibitors A, B, and C have potential advantages in controlling drug doses and toxicity, but their safety and pharmacokinetics still need to be further investigated.

## 4. Materials and Methods

### 4.1. Molecular Docking

Molecular docking was performed using Sybyl-X 2.0 software and MOE software. In the analysis using Sybyl-X 2.0 software, the energy of the compounds was minimized under the Tripos force field. The distance was used as the dielectric function, and the Gasteiger–Hockel atomic charge was applied. The Powell energy gradient method was applied to optimize the lowest energy conformation, and the maximum number of energy optimizations was set to 10,000. The energy convergence criterion was set to 0.001 kcal·mol^−1^, which mimicked the stable conformation of molecules in natural systems. The structure of the PRMT5:MEP50 complex was downloaded from the PDB database and imported into Sybyl-X 2.0 for molecular docking. The semiflexible docking method was used for molecular docking, and the optimized compound and processed protein were docked with the Surflex-Dock module. The total score in Surflex-Dock can reflect the binding affinity between the ligand and the receptor, considering factors such as polarity, hydrophobicity, entropy, and solvation. 

MOE software was used to predict the binding modes of the compounds when binding to PRMT5. The crystal structure of human PRMT5 (PDB ID: 4X61) was retrieved from the protein data bank for ARG and SAM binding site docking [28], and the crystal structure with PDB ID 6UXX was retrieved for allosteric site docking [29]. Hydrogen atoms were added, and the proteins were protonated using Protonate3D. Missing residue sidechains and loops were built. The structures of the compounds were built in MOE 2014 and optimized using the MMFF94x forcefield. The binding sites of SAM and ARG in the structure of 4X61 and the allosteric site in the structure of 6UXX were defined as active sites for molecular docking. The placement method was set to Triangle Matcher and rescored with London dG. The docking conformations were refined with the forcefield and rescored with GBVI/WSA dG. Finally, 30 conformations of each compound were retained, and the conformation with the best S-score was used for structural analyses. In the two methods, a larger absolute value of the docking score indicated a stronger binding ability.

### 4.2. SPR Analysis

A Biacore T200 instrument (GE Healthcare, Little Chalfont, UK) was used for the SPR analyses. The recombinant PRMT5/MEP50 complex (31521, Active Motif, Carlsbad, CA, USA) was immobilized on carboxymethylated dextran CM7 sensor chips (GE Healthcare) using an amine-coupling strategy. A solution of *N*-hydroxysuccinimide and 3-(*N*,*N*-dimethylamino)-propyl-*N*-ethylcarbodiimide (1:1) was used to activate the sensor chip surface. Then, the PRMT5/MEP50 complex was solubilized in acetate buffer and injected in PBS running buffer (10 μL/min) to reach an immobilization level of 20,000 relative units on the CM7 sensor chips. Then, an ethanolamine solution was used to block the surfaces. The binding kinetics of chemicals A, B, and C to the PRMT5/MEP50 complex sensor chip were evaluated in PBS buffer with concentrations ranging from 31.75 to 500 μM. All experiments were performed at 25 °C (30 μL/min). The surfaces of the CM7 sensor chips were then regenerated with two injections of a glycine-HCl solution. Binding sensograms were acquired by subtracting the reference flow cell, and BIA Evaluation Software was used for data analysis.

### 4.3. Cell Culture

The breast cancer cell lines MDA-MB-231, MCF-7, and HCC1937 used in this study were purchased from the American Type Culture Collection (ATCC, Manassas, VA, USA). MDA-MB-231 and HCC1937 cells were maintained in DMEM containing 10% fetal bovine serum and 1% antibiotics, and MCF-7 cells were maintained in DMEM containing 0.2 IU/mL insulin, 10% fetal bovine serum, and 1% antibiotics. Cells were cultured at 37 °C in a 5% CO_2_ incubator. For the Cell Counting Kit-8 (CCK-8) cell proliferation and cytotoxicity experiment, cells were seeded into 96-well plates at 1 × 10^4^ cells/well. For western blotting, cell cycle investigation, and apoptosis detection, cells were seeded into 6-well plates at 1.5 × 10^5^ cells/well. For an EdU incorporation experiment, 2 × 10^4^ cells/well were seeded in a 48-well plate. In the above experiments, the drug treatments were performed on the second day after cell inoculation, and detection was performed 48 h later. Treatments with cisplatin (Selleck, S1166, Houston, TX, USA), doxorubicin (Selleck, S1208), olaparib (Selleck, S1060), and tadalafil (MCE, HY-90009A, Monmouth Junction, NJ, USA) were performed as described in the main text.

### 4.4. Cell Proliferation Assays

CCK-8 (MCE) experiments were performed to assess cell proliferation according to the manufacturer’s instructions. The CCK-8 solution was added to cells in 96-well plates, and incubation was continued for another 1 h at 37 °C. The plates were read at 450 nm with a microplate reader.

EdU assays were performed with a Cell-Light EdU DNA Cell Proliferation Kit (RiboBio, Guangzhou, China). Cells were incubated with a 50 μM EdU solution in the medium for 2 h, fixed with 4% paraformaldehyde (PFA) for 30 min at room temperature and stained with an Apollo staining solution. Images were acquired under a fluorescence microscope (NI-U, Nikon, Tokyo, Japan).

### 4.5. Apoptosis Analysis

For apoptosis analysis, the cells were harvested and washed twice with cold PBS. Afterward, 400 μL of Binding Buffer was added to the cells, followed by the addition of 5 μL of an annexin Ⅴ staining solution and incubation in the dark at room temperature for 15 min. Thereafter, 10 μL of a PI staining solution was added, and incubation was continued on ice for 5 min. The analysis was performed with a FACSCalibur^TM^ flow cytometer.

### 4.6. Western Blotting

Cells were treated with chemicals of different concentrations for 48 h, and cell lysates were obtained with RIPA buffer containing 1 mM PMSF. The protein concentration was measured with a BCA protein quantification kit (Thermo Fisher Scientific, Waltham, MA, USA). Protein samples were separated by SDS/PAGE and then blotted onto polyvinylidene difluoride membranes, which were incubated with 5% skim milk for 1 h. Immunoblotting was performed with an anti-H4R3me2s antibody (1:1000, 61988, Active Motif), anti-H3R8me2s antibody (1:1000, 13939, CST, Danvers, MA, USA), anti-cleaved PARP antibody (1:1000, 5625, CST), and anti-β-actin antibody (1:5000, 3700, CST), followed by incubation with horseradish peroxidase-linked anti-rabbit IgG (1:2000, 7074, CST) or an anti-mouse IgG secondary antibody (1:2000, 7076, CST). The membranes were imaged using a chemiluminescence system (Clinx Science Instruments, Shanghai, China).

### 4.7. Mouse Xenograft Model

All animal experiments were approved by the Animal Experiment Administration Committee of Fourth Military Medical University (Approval Code: 20203285-1, Approval Date: 8 April 2020). MDA-MB-231 cells (5 × 10^6^) were subcutaneously injected into the right flanks of 6-week-old female Balb/c nude mice. Ten days post-inoculation (dpi), the tumor-bearing mice were randomly sorted into different groups. Chemical A (2 mg/kg) were administered by gastric gavage every day from 10 dpi. Olaparib (50 mg/kg) was injected intraperitoneally (i.p.) every day from 10 dpi. Doxorubicin (2 mg/kg) was delivered intravenously once per week from 10 dpi. Cisplatin (2.5 mg/kg) was injected i.p. every three days beginning from 10 dpi. The compounds were formulated according to the product instructions. Tumor size was monitored once every 3 days using a caliper and calculated as π × [d^2^ × D]/6 (d, short diameter; D, long diameter). The tumors were dissected at 31 dpi, fixed with 4% PFA at 4 °C overnight, and then embedded in paraffin using a routine procedure. Paraffin-embedded tissue was stained with anti-Ki67 (1:200, ab15580, Abcam, Cambridge, UK) and anti-γH2A.X (1:400, 9718, CST) antibodies.

### 4.8. Statistical Analysis

Quantitative analysis was performed using Image-Pro Plus 6.0 (Media Cybernetics, Rockville, MD, USA), Fiji v2.0.0 (National Institutes of Health, Bethesda, MD, USA), and FlowJo V7.6.5 (BD Biosciences, Franklin Lakes, NJ, USA) software. Statistical analysis was performed using GraphPad Prism 8.0 software. All quantitative data are presented as the mean ± SEM. Statistical significance was calculated using one-way ANOVA with Tukey’s multiple comparison test. A *p* < 0.05 was considered to indicate statistical significance.

## 5. Conclusions

In summary, we designed and synthesized three new PRMT5 inhibitors through the chemical modification of tadalafil. These three inhibitors have favorable antitumor effects when used alone or in combination with chemotherapeutics. Therefore, these new PRMT5 inhibitors have the potential to become novel molecular targeted drugs that clinically enhance the patients’ tumor cell chemosensitivity, but further study is still needed.

## Figures and Tables

**Figure 1 ijms-23-04806-f001:**
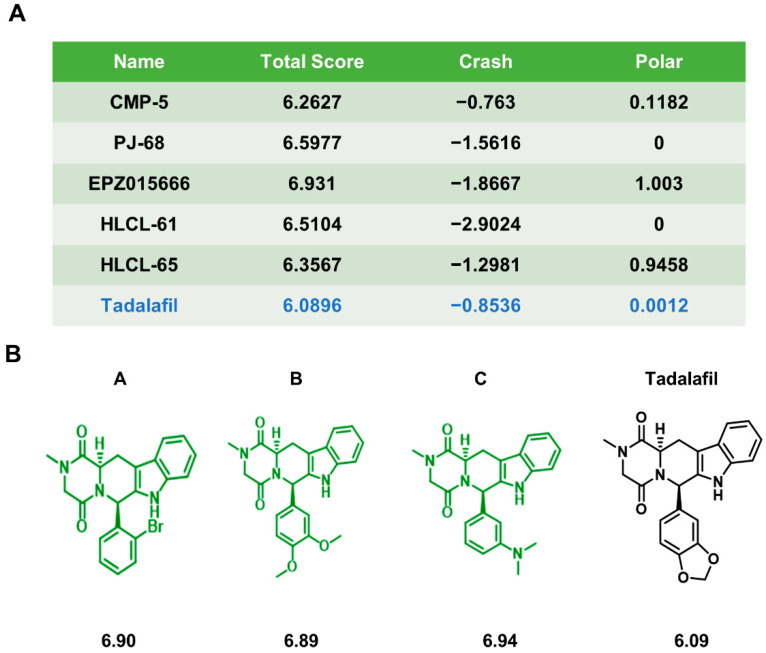
Chemical synthesis of chemicals A, B, and C. (**A**) Molecular docking scores of tadalafil and some existing PRMT5 inhibitors with PRMT5. (**B**) Chemical structures and molecular docking scores of chemicals A, B, C, and tadalafil.

**Figure 2 ijms-23-04806-f002:**
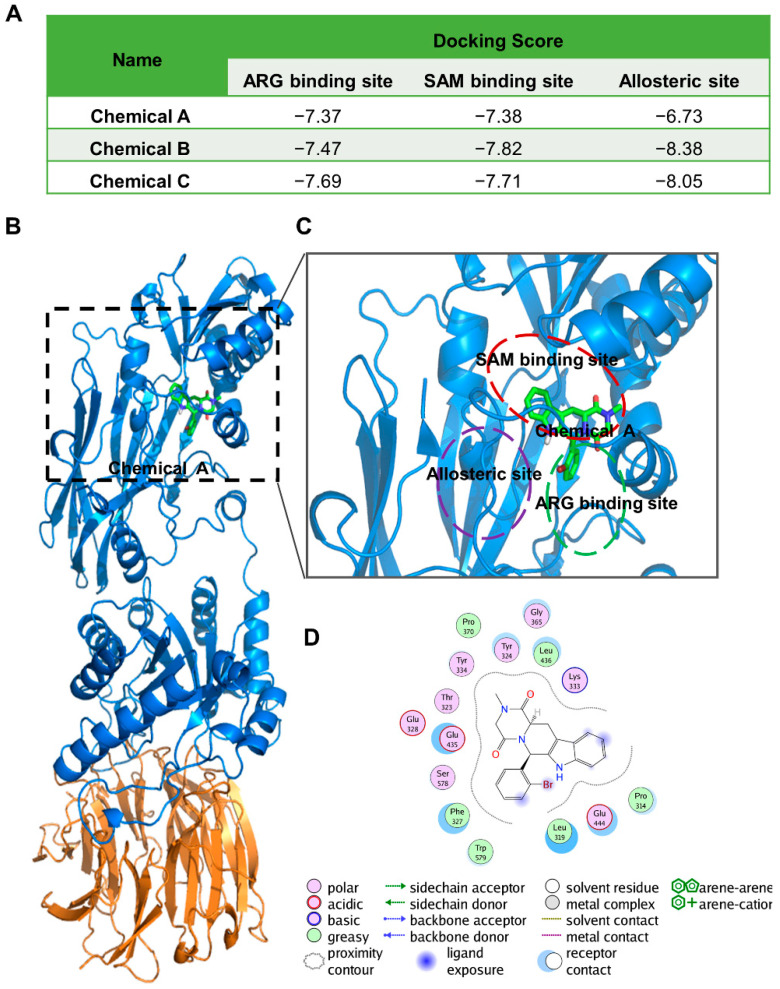
Chemicals A, B, and C could bind to PRMT5 through different binding modes. (**A**) The docking scores of chemicals A, B, and C at the three active sites of PRMT5. (**B**–**D**) Binding interactions of chemical A with human PRMT5. The overall structure of PRMT5:MEP50 complexed with chemical A at the SAM binding site is shown in (**B**). PRMT5 is shown as a blue ribbon, MEP50 is shown as an orange ribbon, and chemical A is shown as green sticks. Detailed binding interactions of chemical A with human PRMT5 are shown in (**C**,**D**).

**Figure 3 ijms-23-04806-f003:**
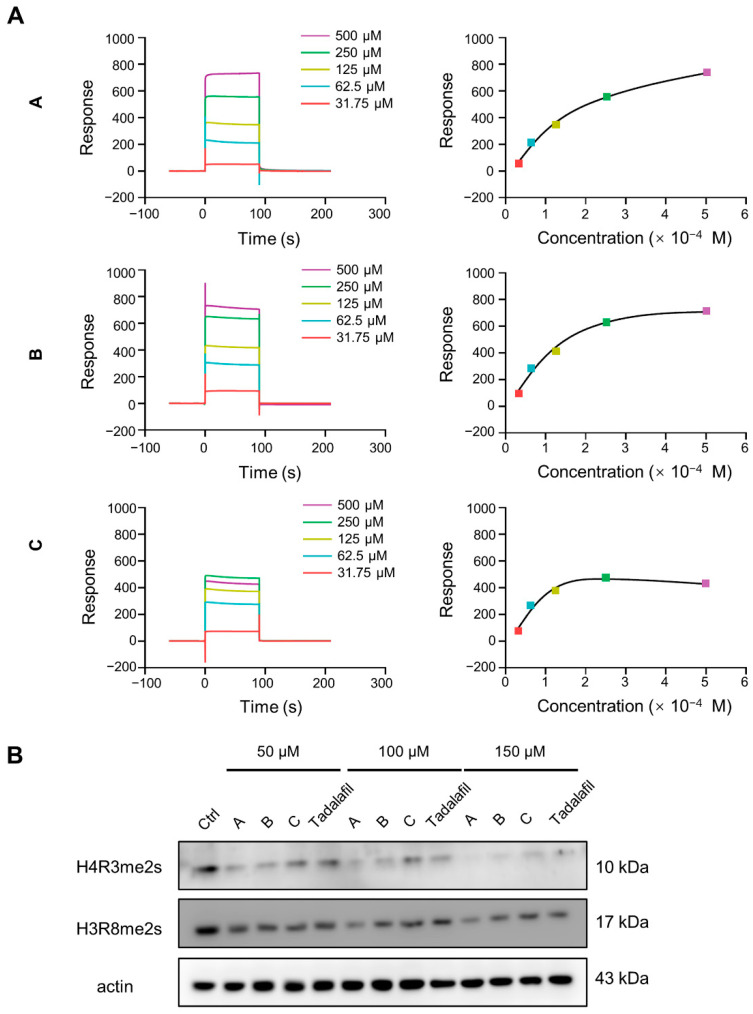
Chemicals A, B, and C bind to PRMT5 and inhibit histone arginine methylation. (**A**) Binding abilities of chemicals A, B, and C to the PRMT5/MEP50 complex detected by Biacore T200. (**B**) MDA-MB-231 cells were treated with the indicated concentrations of chemicals A, B, C, and tadalafil for 48 h, and the global protein levels of H4R3me2s and H3R8me2s were detected by immunoblotting. The picture is representative of two independent experiments.

**Figure 4 ijms-23-04806-f004:**
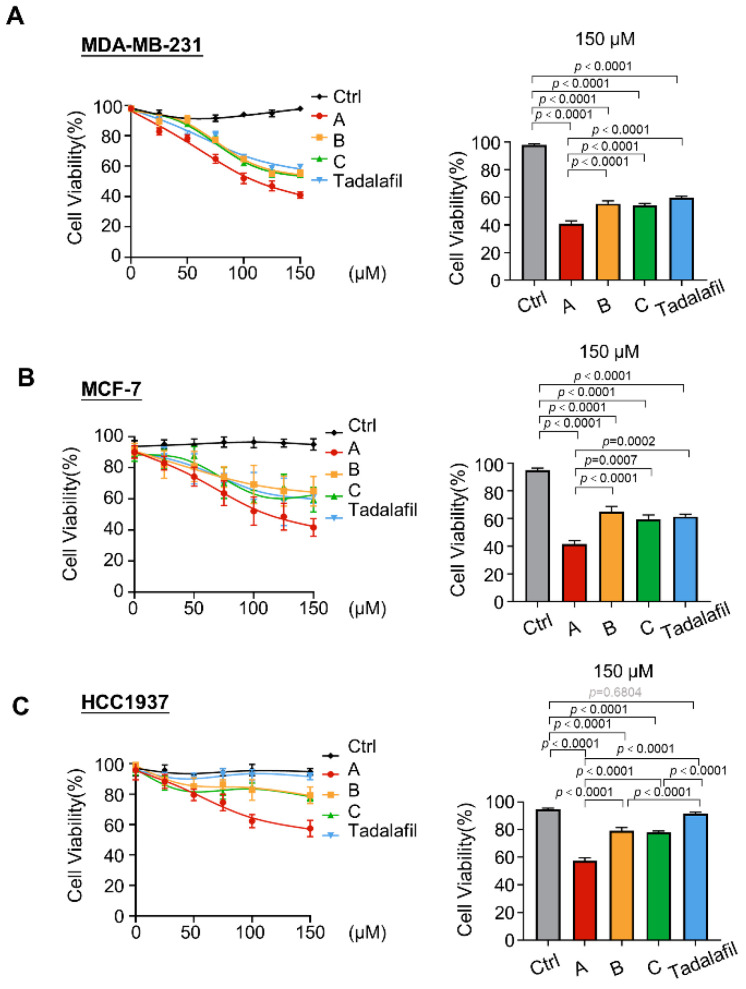
Chemicals A, B, and C inhibit the viability of breast cancer cells. (**A**–**C**) MDA-MB-231 cells (**A**), MCF-7 cells (**B**), and HCC1937 cells (**C**) were treated with the indicated concentrations of chemicals A, B, C, and tadalafil for 48 h, and cell viability was determined by the CCK-8 assay. The cell viability at 150 μM was quantitatively compared (*n* = 6). The line and error bars indicate the mean ± s.e.m.; one-way ANOVA with Tukey’s multiple comparisons test.

**Figure 5 ijms-23-04806-f005:**
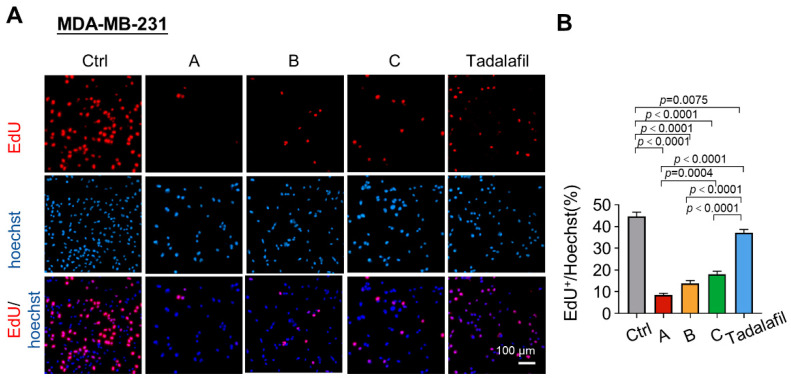
Chemicals A, B, and C suppress the DNA replication of breast cancer cells. MDA-MB-231 cells were treated with chemicals A, B, C, and tadalafil at 150 μM for 48 h, and cell proliferation was detected by EdU assays. (**A**) Representative images of MDA-MB-231 cells treated with chemicals A, B, C, and tadalafil. Scale bar, 100 μm. (**B**) The percentage of EdU-positive (EdU^+^) cells was quantitatively compared (*n* = 8). The line and error bars indicate the mean ± s.e.m.; one-way ANOVA with Tukey’s multiple comparisons test.

**Figure 6 ijms-23-04806-f006:**
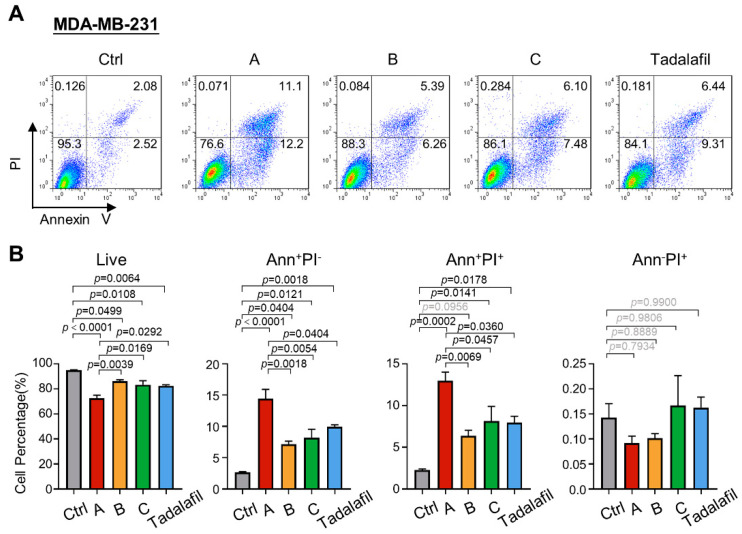
Chemicals A, B, and C promote the apoptosis of MDA-MB-231 cells. MDA-MB-231 cells were treated with chemicals A, B, C, and tadalafil at 150 μM for 48 h, and apoptosis was detected by annexin V/PI staining and FACS. (**A**) Representative plots for the flow cytometric analysis of cell apoptosis. (**B**) Quantification of the cell percentage at different stages of apoptosis (*n* = 3). The line and error bars indicate the mean ± s.e.m.; one-way ANOVA with Tukey’s multiple comparisons test.

**Figure 7 ijms-23-04806-f007:**
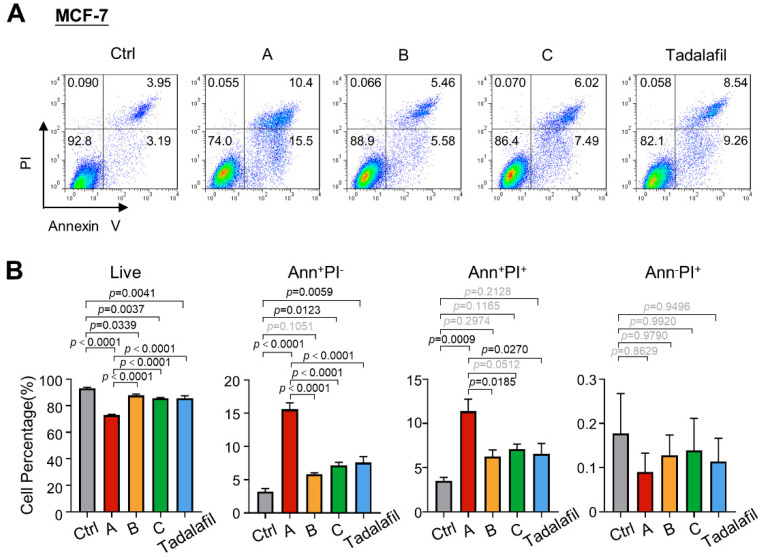
Chemicals A, B, and C promote the apoptosis of MCF-7 cells. MCF-7 cells were treated with chemicals A, B, C, and tadalafil at 150 μM for 48 h, and apoptosis was detected by annexin V/PI staining and FACS. (**A**) Representative plots for the flow cytometric analysis of cell apoptosis. (**B**) Quantification of the cell percentage at different stages of apoptosis (*n* = 3). The line and error bars indicate the mean ± s.e.m.; one-way ANOVA with Tukey’s multiple comparisons test.

**Figure 8 ijms-23-04806-f008:**
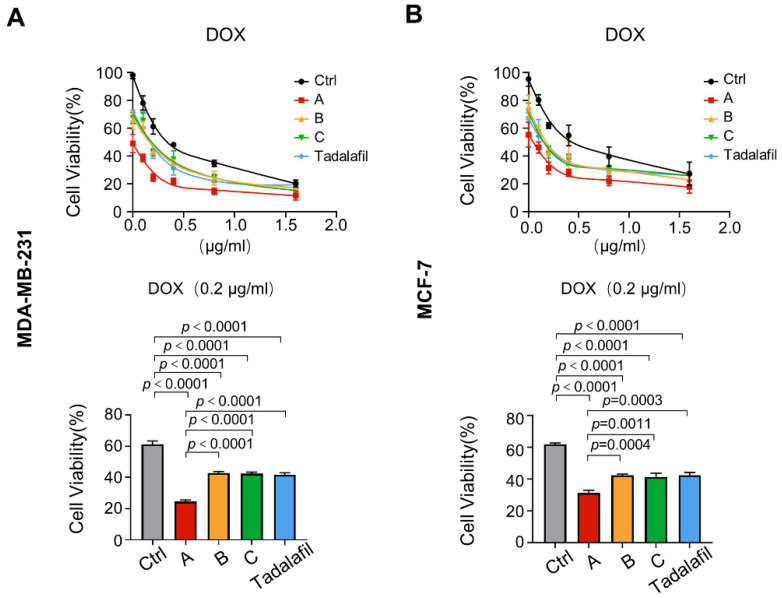
Chemicals A, B, and C enhance the inhibitory effects of doxorubicin on cell viability. (**A**,**B**) MDA-MB-231 (**A**) and MCF-7 cells (**B**) were treated with chemicals A, B, C, and tadalafil at 150 μM in combination with the indicated concentrations of doxorubicin for 48 h. The cell viability at 0.2 μg/mL doxorubicin treatment was quantitatively compared (*n* = 6). The line and error bars indicate the mean ± s.e.m.; one-way ANOVA with Tukey’s multiple comparisons test.

**Figure 9 ijms-23-04806-f009:**
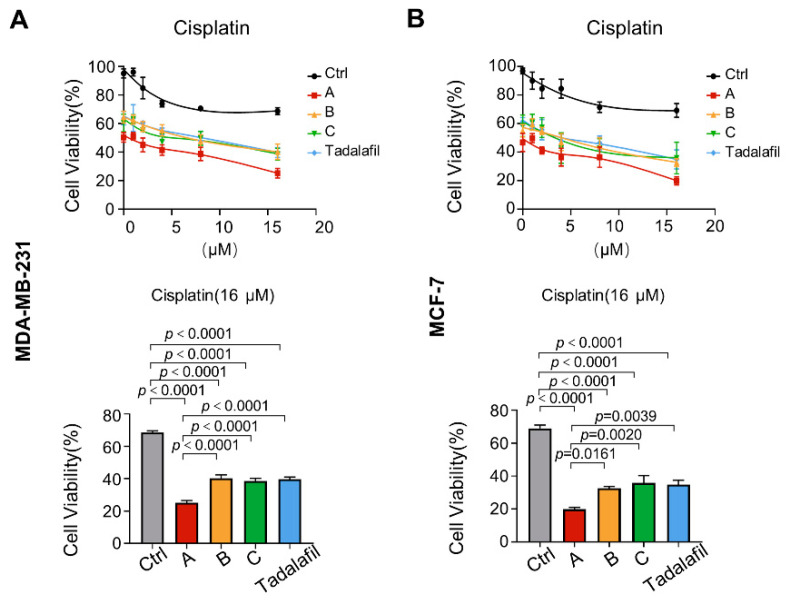
Chemicals A, B, and C enhance the inhibitory effects of cisplatin on cell viability. (**A**,**B**) MDA-MB-231 (**A**) and MCF-7 cells (**B**) were treated with chemicals A, B, C, and tadalafil at 150 μM in combination with the indicated concentrations of cisplatin for 48 h. The cell viability at 16 μM cisplatin treatment was quantitatively compared (*n* = 6). The line and error bars indicate the mean ± s.e.m.; one-way ANOVA with Tukey’s multiple comparisons test.

**Figure 10 ijms-23-04806-f010:**
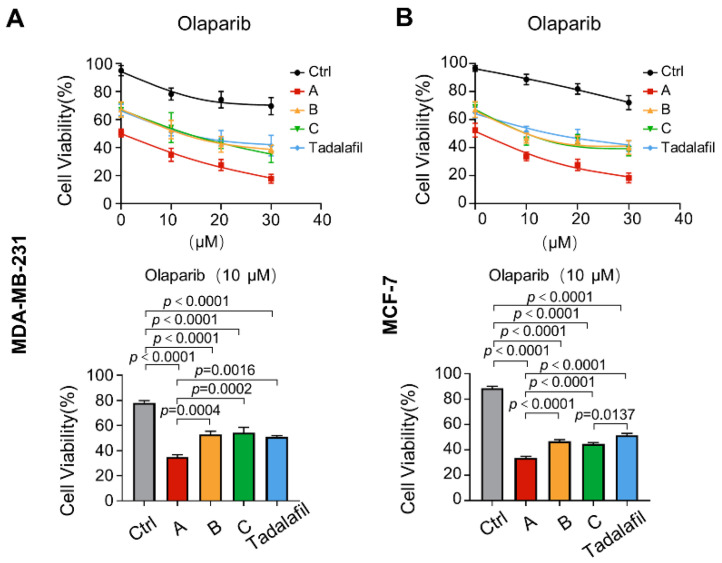
Chemicals A, B, and C enhance the inhibitory effects of olaparib on cell viability. (**A**,**B**) MDA-MB-231 (**A**) and MCF-7 cells (**B**) were treated with chemicals A, B, C, and tadalafil at 150 μM in combination with the indicated concentrations of olaparib for 48 h. The cell viability at 10 μM olaparib treatment was quantitatively compared (*n* = 6). The line and error bars indicate the mean ± s.e.m.; one-way ANOVA with Tukey’s multiple comparisons test.

**Figure 11 ijms-23-04806-f011:**
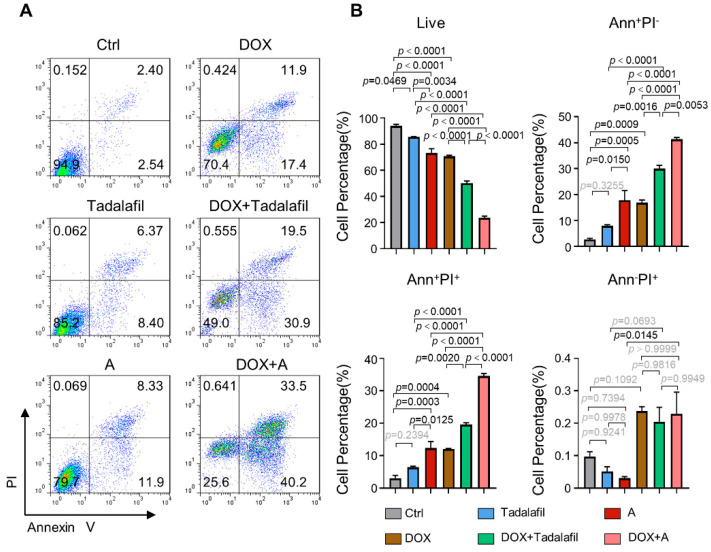
Chemical A improves the pro-apoptotic effects of doxorubicin. MDA-MB-231 cells were treated with 150 μM chemical A and tadalafil in combination with 0.2 μg/mL doxorubicin for 48 h, and apoptosis was detected with annexin V/PI staining and FACS (*n* = 3). (**A**) Representative plots for the flow cytometric analysis of cell apoptosis. (**B**) Quantification of the cell percentage at different stages of apoptosis (*n* = 3). The line and error bars indicate the mean ± s.e.m.; one-way ANOVA with Tukey’s multiple comparisons test.

**Figure 12 ijms-23-04806-f012:**
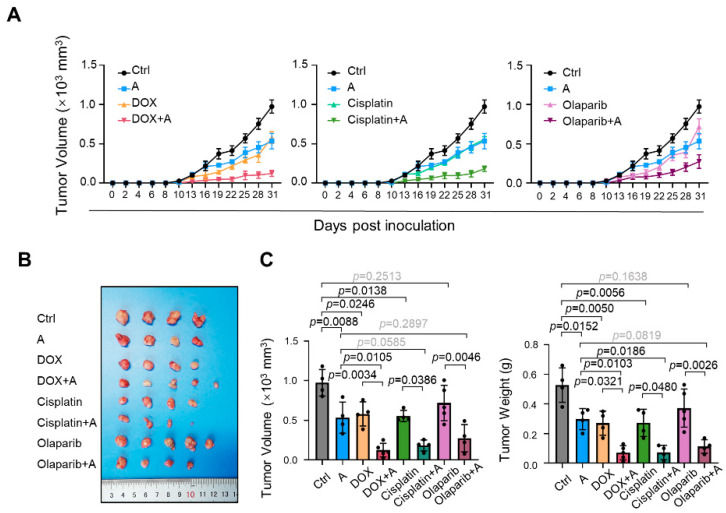
Chemical A enhances the effects of chemotherapy drugs on tumor growth inhibition. Nude mice bearing MDA-MB-231 cells were administered chemical A in combination with doxorubicin, cisplatin, and olaparib from 10 to 31 dpi. (**A**) Tumor growth curves of mice with different treatments are presented. (**B**) Photograph showing the tumors dissected at 31 dpi. (**C**) Tumor weight and tumor volume at 31 dpi were compared. *n* = 4–5. The lines and error bars indicate the means ± s.e.ms.; one-way ANOVA with Tukey’s multiple comparisons test.

**Figure 13 ijms-23-04806-f013:**
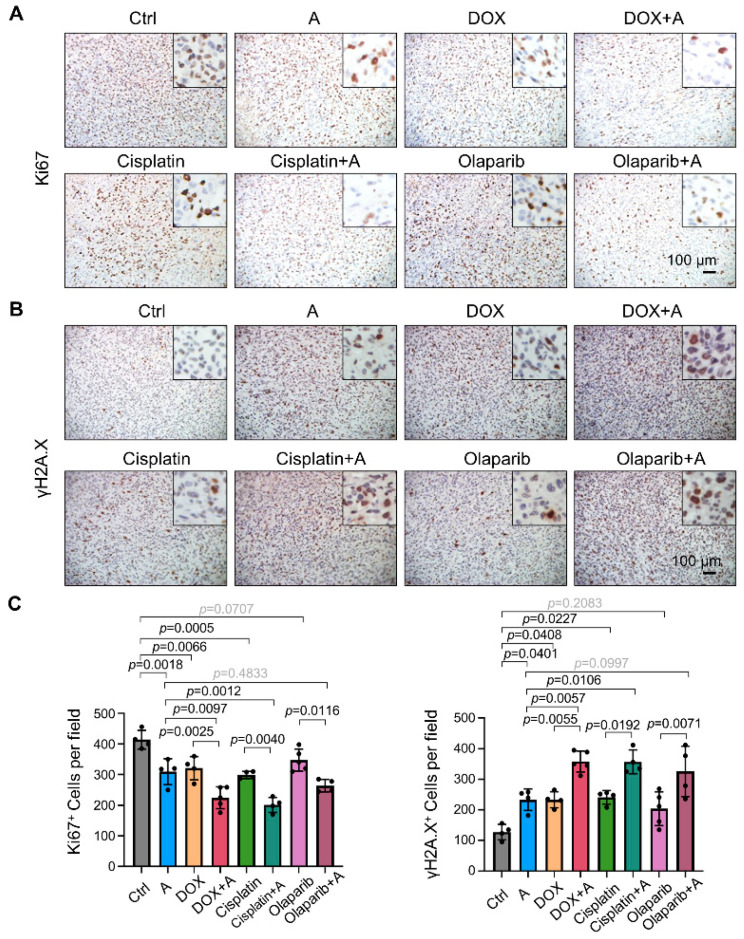
Chemical A enhances the effects of chemotherapeutic drugs on proliferation inhibition and DNA damage induction. Nude mice bearing MDA-MB-231 cells were administered chemical A in combination with doxorubicin, cisplatin, and olaparib from 10 to 31 dpi. (**A**) Representative images showing immunohistochemical staining for Ki67 of tumors dissected at 31 dpi. (**B**) Representative images showing immunohistochemical staining for γH2A. X of tumors dissected at 31 dpi. (**C**) Immunohistochemical staining results in (**A**,**B**) were quantitatively compared. *n* = 4–5. Scale bars, 100 μm. The lines and error bars indicate the means ± s.e.ms.; one-way ANOVA with Tukey’s multiple comparisons test.

## Data Availability

The data presented in this study are available within the article and in the Appendix A.

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
