# Peer review of "Novel Chemicals Derived from Tadalafil Exhibit PRMT5 Inhibition and Promising Activities against Breast Cancer"

_ijms, 2022, doi:10.3390/ijms23094806_

Round 1

Reviewer 1 Report

Dear authors, 

The topic of the paper is interesting and meets the aims and scope of the journal but it has some problems

 However, I have some suggestions for the paper improvement, as follows:

Introduction:

I think that there is a need for reviewing similar researches in other countries  and  you should mention their results. With this way the you  should demonstrate the need of this study.

Methods: Τhe methods described in sufficient detail to understand the approach used and  arrpropiate  statistical tests are applied

Results are detailed and well written 

Discussion and Interpretation are well written and have  reasonable extension of the results

Ethics: i am  not sure it is refferrd appropiately. In this case  I propose to you  to mention exactly if  you have the  proper approval  for this study and mention it in the text 

Reviewer 2 Report

The manuscript submitted by Ziyan et al. analyzed whether the use of various compounds based on the drug Taladafil, either used by themselves or in combination with well established chemotherapeutic agents such as cisplatin, doxorubicin or olaparib, demonstrated synergistic effects in cytotoxicity and apoptosis induction on specific populations of breast cancer cells (MDA-MB-231, MCF-7 and HCC1937). To this end, the authors conducted an experimental, observational study on  groups of cells subjected to the proposed agents. The authors investigated molecular docking scores for the proposed compounds, in vitro cell proliferation and apoptosis analyses, SPR analysis, as well as in-vivo analysis using xenograft murine models. The authors concluded that the newly created PRMT5 inhibitors may be potential candidates for novel therapeutic strategies in the treatment of breast cancer. The main strength of this paper is that it addressed a novel approach to the adjuvant treatment of breast tumors, which may have significant implications for clinical practice.

Title and abstract: The title and abstract are appropriate for the content of the text.

Introduction: The introduction paragraph contains appropriate references and summarizes the present understanding of the discussed topic. In line 50 the authors state that " our studies have shown that the inhibition of PRMT5 can regulate the sensitivity to doxorubicin in the treatment" but provide a single reference. Perhaps this sentence would benefit from revision.

In line 54 the authors use the term "SAM inhibitors". This phrase is somewhat misleading, perhaps the term " SAM-competitive molecules" might better describe the nature of these molecules.

In lines 57-58 the authors state that " However, none of the inhibitors are available on the market. This is a prominent challenge regarding the targeting of PRMT5 for breast cancer treatment." This statement should be revised, since it implies that the paucity of commercially available PRMT5 inhibitors has an influence on the scientific validity of available data for this class of therapeutic agents.

Materials and methods: The description of the experimental design are adequately described. However, this chapter is placed at the end of the paper. Perhaps using the standard format for scientific papers (introduction, materials and methods, results, discussions, conclusions) would have been more appropriate.

Results: Firstly, the results section should follow the materials and methods section, as previously stated. Secondly, the authors did not present the p-values for the ANOVA test, which raises questions concerning the validity of the presented data and the possibility of bias through data dredging. The authors simply state that all p values were included in set categories (<0.05, <0.01 , < 0.001, < 0.0001 or not significant).

Finally, the included figures (especially 4-7) are relatively small and crowded, which makes the data difficult to interpret.

Discussions: The results are adequately discussed in relation to evidence currently available in the literature. However, the final paragraph of this section (line 327-331) would be better presented as a separate conclusions section.

In line 317-319 the authors state that " the use of compounds A, B, and C could enhance the antitumor effect of the PARP inhibitor olaparib on BRCA1-nonmutated cells, such as MDA-MB-231 and MCF-7 cells, which indicated that PARP inhibitors could be effective for both patients with BRCA1 deficiency and patients with normal BRCA1 expression".  Both MDA-MB-231 and MCF-7 are BRCA1 wild-type cell lines, therefore this sentence is somewhat confusing and requires revision in order to clarify the authors point.

 In line 322 the authors use the phrase " early embrionic lethal", which would benefit from revision in order to clarify its meaning.

In addition, perhaps a discussion of the limitations and strengths of the present study would have been appropriate.

A final remark would be the inordinate amount of self citation of the study " [MOLECULAR-THER- 297 APY-D-21-01418, accepted" which is included as an appendix. The study is cited in the introduction, results and discussions paragraph, for a total of 6 times, which is highly inappropriate. This is a major concern and should be revised.

Lastly, while the use of language is mostly sound, there are a few instances there certain points are unclear, making the narrative difficult to follow. A revision of grammar and syntax is required in order to improve the flow and readability of the text.

Author Response

Dear Reviewer:

    We appreciate very much for your thoughtful and constructive comments on our manuscript. We have read and discussed these comments very carefully, and we now make a point-to-point response to answer the questions and  modifications of our manuscript. The revised manuscript are attached in untracked and tracked formats. Please see the attachment.

Sincerely,

Jian Zhang
